# Effects of Essential Oil Blends on In Vitro Apparent and Truly Degradable Dry Matter, Efficiency of Microbial Production, Total Short-Chain Fatty Acids and Greenhouse Gas Emissions of Two Dairy Cow Diets

**DOI:** 10.3390/ani12172185

**Published:** 2022-08-25

**Authors:** Rosetta M. Brice, Peter A. Dele, Kelechi A. Ike, Yasmine A. Shaw, Lydia K. Olagunju, Oluteru E. Orimaye, Kiran Subedi, Uchenna Y. Anele

**Affiliations:** 1Department of Animal Sciences, North Carolina Agricultural and Technical State University, Greensboro, NC 27411, USA; 2Analytical Services Laboratory, College of Agriculture and Environmental Sciences, North Carolina Agricultural and Technical State University, Greensboro, NC 27411, USA

**Keywords:** corn silage, feed efficiency, total mixed ration, plant nutraceuticals, ruminant nutrition

## Abstract

**Simple Summary:**

Livestock accounts for an estimated 80% of total agricultural greenhouse gas emissions, making abatement of greenhouse gas emissions from livestock a high-priority challenge facing animal nutritionists. Mitigating greenhouse gases in ruminants without reducing animal production is desirable both as a strategy to reduce global greenhouse gas emissions and as a way of improving dietary feed efficiency. The inclusion of feed additives in the diets of ruminants can reduce energy losses as methane, which typically reduces animal performance and contributes to greenhouse gas emissions. The present study evaluated the abatement potential of nine essential oil blends to mitigate greenhouse gas emissions. The inclusion of the blends resulted in a reduction in greenhouse gas emissions and in vitro apparent dry matter digestibility with higher values noted for the control treatment. A similar trend was noted for in vitro truly dry matter digestibility with higher values noted in the control treatment. The efficiency of microbial production was greater for the blends. The inclusion of the blends affected the total and molar proportion of volatile fatty acid concentrations. Overall, inclusion of the blends modified the rumen function resulting in improved efficiency of microbial production.

**Abstract:**

The current study evaluated nine essential oil blends (EOBs) for their effects on ruminal in vitro dry matter digestibility (IVDMD), efficiency of microbial production, total short-chain fatty acid concentration (SCFA), total gas, and greenhouse gas (GHG) emissions using two dietary substrates (high forage and high concentrate). The study was arranged as a 2 × 2 × 9 + 1 factorial design to evaluate the effects of the nine EOBs on the two dietary substrates at two time points (6 and 24 h). The inclusion levels of the EOBs were 0 µL (control) and 100 µL with three laboratory replicates. Substrate × EOBs × time interactions were not significant (*p* > 0.05) for total gas and greenhouse gas emissions. The inclusion of EOBs in the diets resulted in a reduction (*p* < 0.001) in GHG emissions, except for EOB1 and EOB8 in the high concentrate diet at 6 h and for EOB8 in the high forage diet at 24 h of incubation. Diet type had no effect on apparent IVDMD (IVADMD) whereas the inclusion of EOBs reduced (*p* < 0.05) IVADMD with higher values noted for the control treatment. The efficiency of microbial production was greater (*p* < 0.001) for EOB treatments except for EOB1 inclusion in the high forage diet. The inclusion of EOBs affected (*p* < 0.001) the total and molar proportion of volatile fatty acid concentrations. Overall, the inclusion of the EOBs modified the rumen function resulting in improved efficiency of microbial production. Both the apparent and truly degraded DM was reduced in the EOB treatments. The inclusion of EOBs also resulted in reduced GHG emissions in both diets, except for EOB8 in the high forage diet which was slightly higher than the control treatment.

## 1. Introduction

Essential oils (EOs) have been known to have varying chemical structures and bioactive compounds as they are extracted from different plants [1]. Essential oils are also known for their antiprotozoal and antimicrobial properties [2,3]. Due to their broad range of antimicrobial activities, EOs are commonly regarded as potential antibiotic replacements [4]. The inclusion of EOs in ruminant diets has been reported to have significant influence in ruminant digestion and fermentation, as well as microbial populations and methanogenesis [5,6,7,8]. The use of EO blends (EOBs) over single EOs has been reported to be more advantageous [3]. Friedman et al. [9] reported that by combining cinnamaldehyde with eugenol, cinnamaldehyde can be preserved from heat-induced degradation. The use of EOBs is often preferred as it is unlikely that a single EO has all the attributes or properties to be as effective as an antibiotic [3]. However, few studies have been carried out to evaluate the effects of different blends of EOs on in vitro rumen fermentation parameters. The effects of EOBs tend to be dependent on the diet or substrate, bioactive compounds, incubation time, and level of inclusion, etc., [5,7]. It is hypothesized that mixing two or more EOs will have synergistic effects to improve animal production. In the present study, seventeen EOs with different chemical structures and bioactives were mixed into nine EOBs. The EOBs were formulated after an extensive review of previous literature and the percent of each oil in the blend was adjusted based on previous results in literature. Our specific objectives were to evaluate the effects of the EOBs on in vitro apparent and truly degraded dry matter, efficiency of microbial production, greenhouse gas emissions, and total and molar proportion of volatile fatty acids.

## 2. Materials and Methods

The cannulated cows used in the present study were maintained under an Institutional Animal Care and Use Committee approved protocol # 21-009.0.

### 2.1. Dietary Substrates

The two dietary substrates were collected from North Carolina A&T State University Farm. The dietary samples consisted of corn silage identified as high forage (HF) and a total mixed ration (TMR) identified as high concentrate (HC; Table 1). The TMR consists of grain products, processed grain byproducts, plant protein products, roughage products, molasses products, and multi-vitamins and minerals supplements. Diets were dried in a forced air oven at 55 °C and milled through a 1 mm sieve and used as substrates for the in vitro batch culture.

### 2.2. Essential Oil Blends

A total of nine EOBs were used for the study. The EOs used in the study were purchased from commercial vendors. The EOBs are as follows: EOB1 (anise, ginger, rosemary, thyme; 5:5:1:5), EOB2 (citronella, clove, lavender, rosemary; 1.25:1.5:2.5:3), EOB3 (anise, cinnamon, oregano, thyme; 1:1:1:1), EOB4 (orange, oregano, garlic, peppermint; 2:2.5:1.5:1), EOB5 (clove, eucalyptus, oregano, peppermint; 3:3:2:2), EOB6 (clove, orange, oregano, thyme; 1:1:1:1), EOB7 (cinnamon, garlic, ginger, lemongrass; 1:1:1:1), EOB8 (eucalyptus, chamomile, cinnamon, tea tree; 2:3:1:1), and EOB9 (lemongrass, lime orange, tea tree; 1:1:1:1). The EOBs were formulated after an extensive review of previous literature and the percent of each oil in the blend was adjusted based on previous results in literature [5,7,8]. The blends were mixed prior to the study and stored at 2 °C degrees.

### 2.3. In Vitro Batch Culture

The batch culture studies were done to evaluate the effect of the EOBs on in vitro dry matter digestibility (IVDMD), efficiency of microbial population (PF, partitioning factor), total and molar proportion of volatile fatty acid concentrations, total gas production, and greenhouse gas emissions (GHG). The studies were arranged as a 2 × 2 × 9 + 1 factorial design to evaluate the effects of the nine EOBs on the two dietary substrates (HF and HC) at two time points (6 and 24 h). The inclusion level of the blends was 100 µL and a control (0 µL). The in vitro incubation procedure was based on the methods described by [10] with some modifications. First, triplicate Ankom (Ankom Technology Corp., Macedon, NY, USA) filter bags were labeled, washed with acetone, and weighed. The bags were filled with approximately 0.5 ± 0.55 g of the feed samples and sealed with an impulse heat sealer. Once sealed, bags were placed into respective 100 mL serum bottles. Following, 100 µL of each EOB was pipetted directly onto the filter bags in triplicate sets. Six bottles with no EOB were also included as a control. Artificial saliva preparation was based on McDougall’s recipe and maintained in a water bath at 39 °C until dispensed into the serum bottles. Ruminal fluid was collected from two ruminally cannulated dairy cows after morning feeding. Their diet consists of 18% protein grain, corn silage, and alfalfa hay daily. The batch culture media was dispensed into the 100 mL glass serum bottles. Each serum bottle received 45 mL of artificial saliva and 15 mL of rumen fluid anaerobically by flushing with carbon dioxide. The bottles were capped with a 14 mm rubber stopper and crimped with an aluminum seal cap. The bottles were incubated on an orbital shaker at 39 °C at a speed of 125 rpm for 6 and 24 h. This process was repeated on a different day for a total of two runs. At the pre-determined time points, headspace gas production was measured at 6 and 24 h post incubation by inserting a 22 mm gauge needle attached to a manometer (VWR International, Randor, PA, USA). 

### 2.4. Estimation of Greenhouse Gases

Methane, ammonia, carbon dioxide, and hydrogen sulfide concentrations were estimated using a portable gas analyzer (Biogas 5000, Landtec, Dexter, MI, USA). The gas analyzer was calibrated per manufacturer’s instruction. An aliquot of gas from the samples were introduced into the analyzer with the aid of a 22 mm gauge needle attached to the end of the inlet tygon tube. Between each sampling, the unit was purged to eliminate any residual gas from the previous sampling.

### 2.5. In Vitro Dry Matter Digestibility

After gas readings, the Ankom bags were removed from the bottles, rinsed, and dried in a 55 °C oven for 48 h. In vitro apparent degradable dry matter (IVADDM) and in vitro true degradable dry matter (IVTDDM) were estimated as described by Anele et al. [11]. 

### 2.6. Microbial Mass Estimation

Microbial mass determination was based on Blümmel and Lebzien [12] with a slight modification. Feed substrates were weighed directly into serum bottles and fermentation was terminated after 24 h. The contents of the bottles were poured into pre-weighed centrifuge tubes. The samples were centrifuged at 20,000× *g* at 4 °C for 15 min. Blanks were also centrifuged and used as a correction factor for residues from the buffered ruminal inoculum. Immediately afterwards, the supernatant was poured off and the samples were placed in a freezer for 24 h. The frozen samples were placed in a freeze dryer (L-200, BUCHI Lyovapor, New Castle, DE, USA) for approximately 48 h. The tubes were then reweighed, and the microbial mass was calculated as: Feed (DM) incubated − [pellet (DM) − blank pellet (DM)]/Feed (DM) incubated

### 2.7. Volatile Fatty Acid Analysis

Volatile fatty acid concentration was based on the protocol of [13]. The samples were analyzed using a gas chromatography (Agilent 7890B GC system/5977B GC-MSD/7693 autosampler, Agilent Technologies, Santa Clara, CA, USA) with a capillary column (Zebron ZB-FFP, Phenomenex Inc., Torrance, CA, USA).

### 2.8. Chemical Analysis

The dietary samples were analyzed for DM (#930.15), N (#954.01), and ether extract (EE; #920.39) according to [14]. Nitrogen was determined using an organic elemental analyzer (2400 CHNS, PerkinElmer, Waltham, MA, USA). Ether extract was determined using the Ankom XT15 (Ankom, Macedon, NY, USA) extractor. Neutral detergent fiber (NDF) and acid detergent fiber (ADF) were analyzed using Ankom 200 Fiber Analyzer (Ankom, Macedon, NY, USA). The NDF content was determined as described by [15] using heat stable α-amylase with sodium sulfite. Acid detergent fiber was determined according to [16] (method 973.18). Acid detergent lignin (ADL) was determined by soaking in concentrated sulfuric acid based on ANKOM Technologies analytical methods.

### 2.9. Statistical Analysis

The data generated were analyzed using the GLM procedure of SAS 9.4 (SAS Inst., Inc., Cary, NC, USA) in a 2 × 2 × 9 + 1 factorial arrangement. Treatment effects and interactions were examined by using the probability of difference (PDIFF) option of the least squares means statement in the MIXED procedure of SAS. The means were separated using Duncan’s multiple comparisons test at *p* < 0.05. Probability values less than 0.001 are expressed as ‘*p* < 0.001’ rather than the actual value.

## 3. Results

The effects of the EOBs and substrates on undegraded DM, IVADDM, IVTDDM, PF, and microbial mass are presented on Table 2. Undegraded DM values were significantly (*p* < 0.001) influenced by EOBs and substrates. The values of the undegraded DM were higher for EOB treatments compared with the control treatment. There was a reduction (*p* < 0.05) in IVADDM with the inclusion of EOBs across the two diets. The IVTDDM values were affected (*p* < 0.001) by both EOBs inclusion and diet with higher values for the control treatment. The PF was influenced by both EOBs inclusion (*p* < 0.001) and the type of substrates (*p* = 0.005). The EOBs treatments recorded greater PF compared with the control treatments across the two substrates. The inclusion of EOBs had no effect (*p* > 0.05) on the microbial mass values across the two substrates.

The effects of EOBs and time on total gas production, methane, carbon dioxide, ammonia, and hydrogen sulfide are presented on Table 3. Substrate × EOBs × time interactions were significant (*p* > 0.001) for total gas production, methane, carbon dioxide, ammonia, and hydrogen sulfide. At 24 h of incubation, the control treatment produced more (*p* < 0.001) gas than the EOB treatments. The GHG values were significantly (*p* < 0.001) reduced by most of the EOBs.

The effects of EOBs and time of incubation on NDF digestibility (NDFD), ADF digestibility (ADFD), and ADL digestibility (ADLD) are presented in Table 4. Degradability of NDF was influenced (*p* < 0.001) with EOBs inclusion and there was significant (*p* = 0.009) EOBs × time interaction. Interactions for EOBs inclusion × substrate × time were not significant for the NDF, ADF, and ADL degradability. Acid detergent fiber and ADL degradability were not affected with EOBs inclusion, but differences (*p* < 0.001) were noted in the substrates with lower values noted for HC diet. 

The effects of EOBs, time, and substrates on the total and molar proportions of VFA production are presented on Table 5. At 6 h of incubation, the inclusion of EOBs reduced the molar proportion of acetate in the HC diet compared with the control but at 24 h of incubation, EOBs 2, 3, 4, 5, and 6 increased the acetate concentration for the HC diet. For the HF diet, inclusion of the EOBs increased acetate concentration at 6 h of incubation for all the EOB treatments except for EOB1. Conversely, at 24 h of incubation, acetate concentration was reduced except for EOBs 2, 3, and 5. The EOB treatments reduced propionate concentrations except for EOBs 1, 4, and 8 at 6 h, and EOB1, 7, and 8 at 24 h of incubation for the HC diet. For the HF diet, EOB inclusion reduced propionate concentration in all but EOB1 and 6 at 6 and 24 h of incubation, respectively. The EOBs increased butyrate concentration of the HC diet except for EOB4 at 24 h of incubation. The EOB7 increased butyrate concentration by 46.7% whereas EOB1 had the lowest increase of 2.8% at 6 h of incubation. At 24 h of incubation, EOB7 increased butyrate concentration of HC by 80.7% whereas EOB4 reduced it by 8.9%. The butyrate concentration of HF at 6 h of incubation was reduced by all the EOBs except for EOBs 2, 3, and 5 and at 24 h of incubation, the butyrate concentration increased with the addition of the EOBs except for EOBs 4, 5, and 6. At 24 h of incubation, EOB7 increased the butyrate concentration of HF by 71.8% and EOB5 reduced it by 17.5%. At 6 h of incubation, the isobutyrate concentration of the HC diet was increased with EOB inclusion, except for EOB2 which was similar to the control treatment. Similarly, isobutyrate concentration was reduced at 24 h of incubation in all the EOB treatments except for EOBs 1, 4, and 8. The EOBs increased isobutyrate concentration in the HF diet at 6 h of incubation except for EOB7 and 9, but at 24 h of incubation, EOB6 increased isobutyrate concentration by 37.5% and the least isobutyrate concentration was by EOBs 2 and 7 which were reduced by 37.5%. The EOBs increased valerate concentration in both substrates when compared to the control treatment. At 6 h of incubation, EOB7 and 8 increased valerate concentration in HC and HF diets by 66.7% and 28.6%, respectively, whereas at 24 h of incubation, EOB3 and 9 increased the valerate concentration of HC and HF by 44.4 and 90.9%, respectively. The isovalerate concentrations in the HC diet at 6 h of incubation were similar (an increase of 33%) for all the EOBs except for EOB5. At 24 h of incubation, the EOBs reduced isovalerate concentration of the HC diet except for EOBs 1 and 8. The EOB8 increased the isovalerate of HF diet at 6 h of incubation by 50% and EOB9 reduced it by 25%. At 24 h of incubation, the isovalerate concentration of the HF diet was reduced by 50% in EOB2 and 9 treatments.

## 4. Discussion

The observed decrease in IVDDM in the present with the inclusion of EOBs was expected as lipids tend to reduce DM digestibility. The decline in digestibility with the inclusion of EOBs could also be associated with a reduction in protozoa population that help to engulf starch in the rumen [17]. A reduction in the protozoa population also has additional advantages as they are mainly responsible for CH_4_ production. For the EOBs, EOB1 was better than the other EOBs in terms of DM digestibility but about 5.6% lower than the control treatment. Higher DM digestibility noted for EOB1 for the HC diet also resulted in a lower undegraded portion when compared with other EOBs, which is an indication that EOB1 had a lower depressing effect on DM digestibility. The EOBs 3 and 6 had the highest undegraded value, which was 33.0% higher than the control treatment and 12.8% higher than EOB1. For the HF diet, the control, EOBs 1 and 8 had similar values for the undegraded material, which implied that these two EOBs did not suppress digestibility as noted for the other EOBs. Variations in the undegraded portions of the substrates reflect their chemical composition with lower values noted for the HC diet due to its lower fiber concentration. Consistent with our result on the effect of the EOBs on the HC diet, [18] reported a decrease in DM digestibility when thymol was supplemented at 500 mg/L in a dual-flow continuous culture system. Several other studies (both in vivo and in vitro studies) have reported similar depression in DM digestibility with inclusion of EOs and their mixtures [17,19,20].

Higher PF values noted for the EOBs with the exception of EOB1 for HC diet showed that the EOBs were able to partition more nutrients into microbial mass which indicates a better efficiency of microbial protein synthesis. The increase in PF in the present study with the inclusion of EOBs, especially EOB4, 5, and 6 for both substrates, could also be as a result of lower gas production and this observation is consistent with previous studies of [21,22] who reported similar results with the inclusion of garlic, rosemary, and thyme essential oils. The PF of a diet indicates the distribution of total organic matter between fermentation gases and microbial biomass and a higher PF value indicates less gas and more microbial biomass generation. After 6 h of incubation, only EOB1 and 8 did not suppress total gas production in the HC diet. The EOB5 treatment reduced total gas production by 96.6% when compared to the control treatment and 96.8% when compared with EOB8 with the highest gas volume. For the HF diet, EOB4, 5, and 6 produced negative net gas volume which implied that they produced less gas than the blanks. The results showed that EOB6 suppressed gas production by 110.3 and 104.7% when compared with the control treatment and EOB8, respectively. After 24 h of incubation, all the EOBs reduced total gas production and compared with the control, EOB5 treatment suppressed total gas by 91.5 and 96.2% for HC and HF, respectively. The present results are consistent with a previous study by [23] who reported a reduction in total gas production with the addition of different doses of essential oils. The trend observed for total gas production in this study, where two EOBs did not suppress the total gas after 6 h of incubation but did after 24 h, could mean that these two EOBs require a time lag before exerting some effects on the rumen microbiome. 

Methane production followed a similar pattern noted for total gas production. Inclusion of EOB5 in the HC diet at 6 h of incubation reduced methane by 98.7 and 99.1% when compared with the control treatment and EOB8, respectively. For HC and HF diets, inclusion of EOBs 4, 5, and 6 reduced methane by 98.1 and 98.6%, 99.9 and 99.1%, and 97.9 and 97.1%, respectively, when compared with the control treatment. These numbers clearly showed that the EOBs vary in their abilities to inhibit methane production. Methane is linked to energy loss, and a decrease in production efficiency and EOs and EOBs could be effective in methane reduction and the modulation of rumen microbiome resulting in improved feed efficiency, and overall productivity in ruminants [24]. Although the EOBs in the present study reduced methane, the intensity of their impact on feed fermentation were varied and dependent on the mixture of EOs used in the blends. Different bioactives present in the different individual EOs could explain the differences in the effects of the EOBs. For instance, EOBs 4, 5, and 6 which had the lowest total gas and methane production had oregano inclusion in the blends either at a higher or equal amount. In some previous studies, oregano being a phenol-based EO has been reported to highly suppress gas and methane production [23,25,26,27]. The above-mentioned EOBs also inhibited other greenhouse gases better than the other EOBs. Carbon dioxide, ammonia, and hydrogen sulphide concentrations followed a similar trend noted for methane concentration where a few of the EOBs did not suppress these gases after 6 h of incubation but reduced their concentrations after 24 h. As stated earlier, this could mean that these EOBs require a time lag before exerting some effects on the rumen microbiome. The only exception to this assumption was EOB8 which produced more GHG than the control treatment. Consistently, net negative production was recorded for the gases in EOBs 4, 5, and 6. The variation noted in ammonia concentration in the present study is an indication that the EOBs had different inhibitory effects on the population of ammonia hyperproducing bacteria [28] and that certain deamination reactions might have been inhibited. The reduction in ammonia concentration with EOBs inclusion is consistent with the report of [23] when origanum oil was used at different inclusion levels. Peppermint (guaiacol) oil inclusion was also reported to reduce ammonia [18].

Results in the current study showed that not all the EOBs suppressed NDFD as EOB1 and 9 had a similar NDFD value with the control treatment for the HC diet after 6 h of incubation. Most of the DM and fiber in a feed or ration are digested by ruminants. As a result, ruminal DM and fiber digestibility are useful indicators for assessing EOs effects [29,30]. In literature, there have been different reports on the effects of EOs on nutrient digestibility by rumen microbes. Some authors reported that EOs have no effect on fiber digestibility [31,32] and others reported that EOs had either beneficial [33,34] or negative [35,36] effects on ruminal digestibility. The EOBs in the present study had different effects on NDFD ranging from no effect to positive and negative effects. The results are varied and suggest that the effects of the EOBs were diet and fermentation time dependent. Higher NDFD with a corresponding higher TVFA value has been reported to be beneficial to ruminants [37]. A decrease in NDFD by some of the EOBs could be that they are rich in phenolic compounds which have been reported to adversely affect NDFD [18,38] by reducing the activities of fibrolytic bacteria. The lack of effect of the EOBs on ADFD is consistent with a previous study by [39] who reported no effect on ADFD when cinnamaldehyde and garlic oils were used in a dual continuous flow system.

Consistent with the varied responses noted in the other variables in the current study, TVFA was reduced with EOBs inclusion for both diets and incubation time points with the exception of EOBs 1 and 9 in the HF diet after 6 h of incubation. Lower TVFA for the HC diet with EOBs inclusion is an indication of a reduction in DM digestibility (DMD) with an exception for EOBs 1, 8, and 9 where DMD was not affected at 6 h of incubation. Contrary to this assumption, the inclusion of EOB5 in the HF diet at 6 h of incubation did not affect DMD despite a reduction in TVFA. EOBs 1 and 9 in the HF diet had no effect on both TVFA and DMD. The lack of change in TVFA of the HF diet at 6 h of incubation with the inclusion of EOB1 could have been considered as positive if it had reduced ammonia and methane concentrations. On the contrary, EOB9 reduced both ammonia and methane concentrations. The lack of effect on TVFA concentration could be considered favorable if it was accompanied by other changes such as lower ammonia and methane concentrations, or a shift in VFA molar proportions [37]. Van Soest [40] reported that the production of VFA during rumen microbial fermentation is a major source of energy for ruminants. Therefore, the reduction in TVFA for some of the EOBs without a concomitant increase in microbial mass may not be ideal. The increase in acetate concentration after 24 h of incubation in EOBs 2, 3, 4, 5, and 6 in the HC diet is viewed negatively as an increase in molar proportion of acetate is associated with an increase in methane concentration. The reduction in the molar proportion of acetate with EOBs inclusion in the HC diet at 6 h of incubation is considered advantageous [41], especially if the EOBs increased the molar proportion of propionate. At 24 h of incubation, the EOB7 inclusion in the HC diet, reduced the acetate proportion and increased the propionate proportion and was able to reduce methane by 94.8%. For the HF diet, EOB7 reduced the propionate proportion which shows that its effect could be substrate or diet dependent although the methane concentration was reduced by 93.8%. The increased propionate concentration noted for EOB1 resulted in a corresponding decrease in acetate and butyrate concentrations for both diets at 6 h of incubation but did not affect methane as well as other greenhouse gases. This could mean that EOB1 has less potential to inhibit methane at the inclusion dose of 100 µL. It is well documented that an increase in the molar proportion of propionate can have a negative influence on methanogenesis by increasing H_2_ competition [42]. The effect of EOB1 at 6 h of incubation on the total and molar proportion of VFA is consistent with a previous study by [39]. At 24 h of incubation, a reduction in propionate proportion in the HF diet by EOB1 is consistent with the report of [5] that variations in the total and molar proportion of VFA are diet and fermentation time dependent. The implication here could be that the rumen microbes adjusted to the impacts of EOB1 over an extended period (from 6 to 24 h). 

## 5. Conclusions

Overall, the EOBs in the present study had different effects on in vitro fermentation characteristics. The inclusion of the EOBs modified the rumen function resulting in improved efficiency of microbial production. Both apparent and truly degraded DM was reduced in the EOB treatments. The inclusion of EOBs also resulted in reduced GHG emissions in both diets, except for EOB8 in the high forage diet which was slightly higher than the control treatment. The observed differences in their effects show that they could be used as feed additives to improve livestock production. Future directions include using different doses of the EOBs as well as examining their effects on microbial diversity.

## Figures and Tables

**Table 1 animals-12-02185-t001:** Chemical composition (% dry matter) of the basal substrate used in ruminal in vitro incubations (*n* = 3).

	High Forage	High Concentrate
Dry matter	96.2	95.8
Crude protein	6.72	16.6
Ether extract	5.95	5.60
Neutral detergent fiber	44.7	32.0
Acid detergent fiber	21.3	13.7
Acid detergent lignin	1.67	2.86

**Table 2 animals-12-02185-t002:** Effects of the essential oil blends (EOBs) and substrates on some fermentation parameters (*n* = 6).

Substrate	Trt	Undegraded	IVADDM ^1^	IVTDDM	PF	Mmass (g/kg DM)
HC ^2^	Control	0.106 ^d^	0.460 ^a^	0.790 ^a^	1.42 ^f–h^	0.166
	EOB1	0.125 ^c^	0.348 ^a^	0.746 ^b^	1.73 ^d–g^	0.197
	EOB2	0.137 ^bc^	0.354 ^a^	0.729 ^bc^	2.13 ^de^	0.189
	EOB3	0.141 ^b^	0.318 ^a^	0.722 ^c^	2.35 ^d^	0.204
	EOB4	0.138 ^b^	0.327 ^a^	0.725 ^bc^	4.13 ^ab^	0.201
	EOB5	0.137 ^bc^	0.321 ^a^	0.727 ^bc^	4.01 ^b^	0.202
	EOB6	0.141 ^b^	0.183 ^a^	0.720 ^c^	3.58 ^bc^	0.271
	EOB7	0.136 ^bc^	0.240 ^a^	0.728 ^bc^	2.13 ^de^	0.242
	EOB8	0.131 ^bc^	0.269 ^a^	0.737 ^bc^	1.63 ^f–h^	0.232
	EOB9	0.137 ^bc^	0.221 ^a^	0.725 ^bc^	1.62 ^f–h^	0.249
HF	Control	0.179 ^a^	0.488 ^a^	0.647 ^d^	1.10 ^gh^	0.141
	EOB1	0.179 ^a^	0.246 ^a^	0.640 ^d^	1.00 ^h^	0.196
	EOB2	0.189 ^a^	0.218 ^a^	0.631 ^d^	1.76 ^d–g^	0.212
	EOB3	0.186 ^a^	0.315 ^a^	0.628 ^d^	2.15 ^de^	0.231
	EOB4	0.190 ^a^	0.250 ^a^	0.626 ^d^	3.27 ^c^	0.191
	EOB5	0.188 ^a^	−0.402 ^b^	0.628 ^d^	4.65 ^a^	0.208
	EOB6	0.185 ^a^	0.232 ^a^	0.633 ^d^	3.71 ^bc^	0.203
	EOB7	0.188 ^a^	0.203 ^a^	0.633 ^d^	1.89 ^d–f^	0.223
	EOB8	0.178	0.302 ^a^	0.641 ^d^	1.10 ^gh^	0.167
	EOB9	0.185 ^a^	0.240 ^a^	0.631 ^d^	1.42 ^f–h^	0.191
SEM		0.003	0.031	0.005	0.109	0.009
Trt		<0.001	0.047	<0.001	<0.001	0.811
Sub		<0.001	0.104	<0.001	0.005	0.333
Trt × sub		<0.001	0.026	<0.001	<0.001	0.967

^1^ IVADDM, In vitro apparent degradable dry matter; IVTDDM, In vitro true degradable dry matter; PF, Partitioning factor; Mmass, Microbial mass. ^2^ HC, High concentrate; HF, High forage; SEM, Standard error of means; Trt, Treatment; Sub, Substrate. ^a–h^ Means with different superscripts within the same column differ, *p* < 0.05.

**Table 3 animals-12-02185-t003:** Effects of the essential oil blends (EOBs) and time on total gas production, methane, carbon dioxide, ammonia, and hydrogen sulfide (*n* = 6).

		Gas (mL/g DM)		Methane (mg/g DM)		Carbon Dioxide (mg/g DM)		Ammonia (mmol/g DM)		HydroS ^1^ (mmol/g DM)	
Time	Trt	HC ^2^	HF	HC	HF	HC	HF	HC	HF	HC	HF
6 h	Control	82.1 ^hij^	82.2 ^hij^	3.13 ^g–k^	4.14 ^g–j^	21.1 ^g–l^	26.4 ^g–j^	68.4 ^g–o^	130 ^efg^	222 ^f–j^	513 ^f^
	EOB1	84.6 ^g–j^	83.4 ^hij^	3.32 ^g–k^	3.85 ^g–k^	25.2 ^g–j^	24.7 ^g–k^	69.2 ^g–o^	127 ^e–h^	236 ^f–j^	386 ^f–j^
	EOB2	51.2 ^i–m^	42.1 ^j–n^	1.73 ^g–k^	1.46 ^h–k^	15.6 ^h–o^	13.4 ^h–o^	25.4 ^k–o^	56.5 ^g–o^	58.0 ^hij^	119 ^f–j^
	EOB3	31.5 ^k–o^	27.4 ^k–o^	0.94 ^ijk^	0.76 ^ijk^	9.17 ^i–o^	8.48 ^j–o^	12.6 ^l–o^	23.7 ^k–o^	27.7 ^hij^	44.4 ^hij^
	EOB4	6.83 ^no^	−0.51 ^no^	0.07 ^k^	−0.01 ^k^	1.86 ^mno^	−0.19 ^o^	7.61 ^mno^	−0.79 ^no^	21.4 ^hij^	−8.39 ^ij^
	EOB5	2.77 ^no^	−1.71 ^o^	0.04 ^k^	−0.01 ^k^	0.85 ^no^	−0.61 ^o^	2.60 ^mno^	−2.46 ^o^	3.35 ^ij^	−12.9 ^ij^
	EOB6	20.6 ^l–o^	−4.25 ^o^	0.27 ^jk^	−0.03 ^k^	6.68 ^k–o^	−1.37 ^o^	11.0 ^mno^	−4.41 ^o^	30.2 ^hij^	−14.6 ^j^
	EOB7	57.0 ^i–m^	56.7 ^i–m^	0.64 ^ijk^	0.63 ^ijk^	19.7 ^h–m^	19.0 ^h–n^	64.4 ^g–o^	97.9 ^e–l^	375 ^f–j^	416 ^f–h^
	EOB8	86.2 ^ghi^	90.8 ^ghi^	4.34 ^ghi^	3.74 ^g–k^	28.4 ^gh^	27.0 ^ghi^	84.9 ^f–n^	123 ^e–i^	281 ^f–j^	363 ^f–j^
	EOB9	67.9 ^ijk^	58.2 ^i–l^	3.46 ^g–k^	2.12 ^g–k^	22.1 ^g–l^	14.0 ^h–o^	40.7 ^h–o^	43.2 ^h–o^	103 ^g–j^	81.8 ^g–j^
24 h	Control	397 ^ab^	403 ^a^	24.1 ^ab^	23.4 ^ab^	111 ^ab^	109 ^ab^	295 ^c^	403 ^ab^	1530 ^cde^	2156 ^ab^
	EOB1	336 ^c^	355 ^bc^	19.3 ^cd^	21.0 ^bc^	85.5 ^cd^	93.6 ^bc^	263 ^c^	398 ^b^	1140 ^e^	1870 ^bc^
	EOB2	184 ^f^	179 ^f^	11.4 ^f^	10.7 ^f^	57.9 ^e^	53.6 ^ef^	89.2 ^e–m^	114 ^e–j^	252 ^f–j^	270 ^f–j^
	EOB3	127 ^g^	123 ^gh^	5.58 ^g^	4.87 ^gh^	39.4 ^fg^	39.1 ^fg^	31.7 ^j–o^	37.4 ^i–o^	94.1 ^g–j^	119 ^f–j^
	EOB4	53.5 ^i–m^	27.7 ^k–o^	0.45 ^ijk^	0.32 ^jk^	15.4 ^h–o^	9.65 ^i–o^	43.4 ^h–o^	31.8 ^j–o^	291 ^f–j^	186 ^f–j^
	EOB5	33.9 ^k–o^	15.3 ^mno^	0.33 ^jk^	0.20 ^k^	11.5 ^h–o^	5.37 ^l–o^	30.7 ^j–o^	39.7 ^i–o^	162 ^f–j^	137 ^f–j^
	EOB6	66.2 ^ijk^	66.1 ^ijk^	0.59 ^ijk^	0.67 ^ijk^	20.0 ^h–m^	22.7 ^g–l^	23.9 ^k–o^	39.4 ^i–o^	248 ^f–j^	245 ^f–j^
	EOB7	194 ^f^	204 ^ef^	1.24 ^h–k^	1.46 ^h–k^	60.3 ^e^	69.2 ^de^	172 ^de^	240 ^cd^	2077 ^ab^	1785 ^bcd^
	EOB8	275 ^d^	365 ^abc^	16.7 ^de^	25.9 ^a^	88.1 ^c^	114 ^a^	245 ^cd^	486 ^a^	1440 ^de^	2301 ^a^
	EOB9	220 ^ef^	239 ^de^	12.5 ^f^	14.3 ^ef^	59.7 ^e^	64.7 ^e^	100 ^e–k^	164 ^d–f^	393 ^f–i^	470 ^fg^
SEM		15.319	15.319	1.408	1.408	6.643	6.643	31.119	31.119	145.72	145.72
Trt		<0.001	<0.001	<0.001	<0.001	<0.001	<0.001	<0.001	<0.001	<0.001	<0.001
Sub		0.839	0.839	0.484	0.484	0.809	0.809	<0.001	<0.001	0.050	0.050
Trt × sub		0.192	0.192	0.427	0.427	0.811	0.811	0.010	0.010	0.007	0.007
Trt × *t*		<0.001	<0.001	<0.001	<0.001	<0.001	<0.001	<0.001	<0.001	<0.001	<0.001
Trt × sub × *t*		<0.001	<0.001	<0.001	<0.001	<0.001	<0.001	<0.001	<0.001	<0.001	<0.001

^1^ HydroS, Hydrogen sulfide; DM, dry matter. ^2^ HC, High concentrate; HF, High forage; SEM, Standard error of means; Trt, Treatment; Sub, Substrate; *t*, time. ^a–o^ Means with different superscripts within the same column differ, *p* < 0.05.

**Table 4 animals-12-02185-t004:** The effects of the essential oil blends (EOBs) and time of incubation on NDFD (%), ADFD (%), and ADLD (%) (*n* = 6).

		NDFD ^1^		ADFD		ADLD	
Time	Trt	HC ^2^	HF	HC	HF	HC	HF
6 h	Control	44.3	35.0	53.9	45.8	7.49	13.2
	EOB1	44.4	33.9	54.5	41.7	11.7	18.5
	EOB2	43.1	34.1	52.3	41.6	4.53	15.2
	EOB3	42.6	33.9	53.4	40.7	11.6	16.1
	EOB4	43.0	27.8	53.3	33.0	6.24	14.6
	EOB5	36.2	34.9	43.1	42.3	3.77	15.1
	EOB6	43.0	33.8	53.8	40.7	6.63	16.5
	EOB7	41.9	34.0	44.5	34.7	5.10	15.2
	EOB8	43.0	29.5	47.5	35.4	6.40	15.0
	EOB9	44.6	36.4	51.4	41.0	4.38	16.9
24 h	Control	64.4	40.2	52.5	38.3	6.76	14.2
	EOB1	55.7	46.3	52.9	45.3	6.34	16.6
	EOB2	51.0	38.8	52.4	42.7	4.23	16.3
	EOB3	48.5	39.6	53.5	43.8	5.03	17.2
	EOB4	47.4	36.2	52.1	43.9	8.79	18.1
	EOB5	48.6	37.7	52.5	42.1	5.03	15.5
	EOB6	49.3	38.5	53.7	42.2	4.81	15.6
	EOB7	54.2	42.9	51.3	43.5	9.75	18.8
	EOB8	59.0	43.6	52.4	43.3	3.99	17.8
	EOB9	50.7	40.0	52.3	41.3	3.65	15.5
SEM		0.61	0.61	0.605	0.605	0.463	0.463
Trt		<0.001	<0.001	0.385	0.385	0.289	0.289
Sub		<0.001	<0.001	<0.001	<0.001	<0.001	<0.001
Trt × sub		0.063	0.063	0.888	0.888	0.762	0.762
Trt × *t*		0.009	0.009	0.239	0.239	0.293	0.293
Trt × sub × *t*		0.204	0.204	0.544	0.544	0.919	0.919

^1^ NDFD, Neutral detergent fiber degradability; ADFD, Acid detergent fiber degradability; ADL, Acid detergent lignin degradability; ^2^ HC, High concentrate; HF, High forage; SEM, Standard error of means; Trt, Treatment; Sub, Substrate; *t*, time.

**Table 5 animals-12-02185-t005:** Effects of the essential oil blends (EOBs), time, and substrates on total and molar proportion of volatile fatty acids (VFA, mM) production (*n* = 6).

Time	Trt	Total VFA		Acetate		Propionate		Butyrate		Isobutyrate		Valeric		Isovaleric	
		HC ^1^	HF	HC	HF	HC	HF	HC	HF	HC	HF	HC	HF	HC	HF
6 h	Control	42.4 ^e–h^	31.5 ^j–o^	0.715 ^abc^	0.665 ^c–h^	0.157 ^f–k^	0.186 ^a–e^	0.107 ^mn^	0.123 ^k–n^	0.006 ^e^	0.007 ^d^	0.012 ^g^	0.014 ^f^	0.003 ^d^	0.004 ^c^
	EOB1	36.6 ^h–k^	32.5 ^j–n^	0.682 ^b–f^	0.647 ^e–j^	0.183 ^a–f^	0.208 ^a^	0.110 ^lmn^	0.115 ^k–n^	0.007 ^d^	0.008 ^c^	0.014 ^f^	0.016 ^d–f^	0.004 ^c^	0.005 ^b^
	EOB2	29.7 ^l–r^	24.5 ^p–t^	0.682 ^b–f^	0.670 ^c–h^	0.154 ^g–k^	0.174 ^c–h^	0.139 ^g–m^	0.129 ^i–n^	0.006 ^e^	0.007 ^d^	0.014 ^f^	0.015 ^ef^	0.004 ^c^	0.004 ^c^
	EOB3	26.0 ^n–t^	23.2 ^r–u^	0.693 ^b–e^	0.680 ^b–f^	0.140 ^jkl^	0.162 ^e–j^	0.142 ^g–l^	0.130 ^i–n^	0.007 ^d^	0.008 ^c^	0.015 ^ef^	0.016 ^d–f^	0.004 ^c^	0.004 ^c^
	EOB4	23.3 ^r–u^	21.1 ^s–v^	0.681 ^b–f^	0.67 ^c–g^	0.161 ^e–j^	0.180 ^b–f^	0.129 ^i–n^	0.119 ^k–n^	0.007 ^d^	0.008 ^c^	0.017 ^c–f^	0.016 ^d–f^	0.004 ^c^	0.004 ^c^
	EOB5	25.7 ^o–t^	21.6 ^s–v^	0.707 ^bcd^	0.669 ^c–h^	0.142 ^jkl^	0.177 ^b–g^	0.125 ^j–n^	0.125 ^j–n^	0.007 ^d^	0.008 ^c^	0.015 ^ef^	0.017 ^c–f^	0.003 ^d^	0.004 ^c^
	EOB6	25.2 ^o–t^	21.9 ^s–v^	0.685 ^b–f^	0.671 ^c–g^	0.151 ^h–k^	0.179 ^b–g^	0.137 ^h–m^	0.121 ^k–n^	0.007 ^d^	0.009 ^b^	0.015 ^ef^	0.016 ^d–f^	0.004 ^c^	0.004 ^c^
	EOB7	27.2 ^m–t^	30.3 ^k–q^	0.658 ^d–i^	0.728 ^ab^	0.154 ^g–k^	0.133 ^kl^	0.157 ^f–j^	0.114 ^k–n^	0.007 ^d^	0.006 ^e^	0.020 ^abc^	0.015 ^ef^	0.004 ^c^	0.004 ^c^
	EOB8	33.8 ^j–m^	28.9 ^l–r^	0.676 ^c–f^	0.673 ^c–g^	0.172 ^d–i^	0.172 ^d–i^	0.123 ^k–n^	0.120 ^k–n^	0.008 ^c^	0.010 ^a^	0.016 ^d–f^	0.018 ^b–e^	0.004 ^c^	0.006 ^a^
	EOB9	26.4 ^n–t^	32.7 ^j–n^	0.696 ^b–e^	0.758 ^a^	0.132 ^kl^	0.121 ^lm^	0.147 ^g–k^	0.100 ^n^	0.007 ^d^	0.006 ^e^	0.015 ^ef^	0.012 ^g^	0.004 ^c^	0.003 ^d^
24 h	Control	57.6 ^a^	52.7 ^abc^	0.620 ^h–l^	0.614 ^i–l^	0.195 ^a–d^	0.191 ^a–d^	0.158 ^f–i^	0.171 ^fg^	0.008 ^c^	0.008 ^c^	0.014 ^f^	0.011 ^h^	0.004 ^c^	0.004 ^c^
	EOB1	55.1 ^ab^	47.9 ^cde^	0.587 ^kl^	0.609 ^i–l^	0.200 ^abc^	0.182 ^a–f^	0.185 ^ef^	0.183 ^ef^	0.008 ^c^	0.007 ^d^	0.016 ^d–f^	0.014 ^f^	0.004 ^c^	0.004 ^c^
	EOB2	35.5 ^i–l^	35.2 ^i–l^	0.640 ^f–j^	0.625 ^g–l^	0.087 ^n^	0.097 ^mn^	0.246 ^c^	0.253 ^c^	0.005 ^f^	0.005 ^f^	0.020 ^abc^	0.018 ^b–e^	0.002 ^e^	0.002 ^e^
	EOB3	30.8 ^k–p^	20.7 ^tuv^	0.626 ^g–l^	0.625 ^g–l^	0.094 ^n^	0.140 ^jkl^	0.250 ^c^	0.204 ^de^	0.005 ^f^	0.008 ^c^	0.022 ^a^	0.020 ^abc^	0.003 ^d^	0.004 ^c^
	EOB4	21.0 ^s–v^	17.3 ^uv^	0.681 ^b–f^	0.610 ^i–l^	0.147 ^ijk^	0.189 ^a–d^	0.145 ^g–k^	0.169 ^f–h^	0.008 ^c^	0.009 ^b^	0.016 ^d–f^	0.019 ^a–d^	0.003 ^d^	0.004 ^c^
	EOB5	23.8 ^q–u^	17.1 ^uv^	0.647 ^e–j^	0.645 ^e–j^	0.139 ^jkl^	0.185 ^a–e^	0.186 ^ef^	0.141 ^g–l^	0.007 ^d^	0.010 ^a^	0.017 ^c–f^	0.015 ^ef^	0.003 ^d^	0.004 ^c^
	EOB6	27.7 ^m–s^	16.1 ^v^	0.636 ^f–k^	0.605 ^jkl^	0.119 ^lm^	0.198 ^a–d^	0.219 ^d^	0.164 ^f–h^	0.006 ^e^	0.011 ^a^	0.018 ^b–e^	0.019 ^a–d^	0.003 ^d^	0.004 ^c^
	EOB7	43.3 ^efg^	37.6 ^g–j^	0.487 ^m^	0.521 ^m^	0.201 ^ab^	0.148 ^ijk^	0.285 ^ab^	0.304 ^a^	0.005 ^f^	0.005 ^f^	0.020 ^abc^	0.020 ^abc^	0.002 ^e^	0.003 ^d^
	EOB8	50.2 ^bcd^	46.1 ^d–f^	0.582 ^l^	0.610 ^i–l^	0.200 ^abc^	0.180 ^b–f^	0.189 ^ef^	0.183 ^ef^	0.008 ^c^	0.009 ^b^	0.017 ^c–f^	0.014 ^f^	0.004 ^c^	0.004 ^c^
	EOB9	40.8 ^f–i^	37.6 ^g–j^	0.583 ^l^	0.602 ^jkl^	0.106 ^mn^	0.104 ^mn^	0.284 ^ab^	0.266 ^bc^	0.005 ^f^	0.005 ^f^	0.020 ^abc^	0.021 ^ab^	0.002 ^e^	0.002 ^e^
SEM		1.02	1.02	0.005	0.005	0.003	0.003	0.005	0.005	0.0009	0.0009	0.0002	0.0002	0.0002	0.0002
Trt		<0.001	<0.001	<0.001	<0.001	<0.001	<0.001	<0.001	<0.001	<0.001	<0.001	<0.001	<0.001	<0.001	<0.001
Sub		<0.001	<0.001	0.491	0.491	<0.001	<0.001	<0.001	<0.001	<0.001	<0.001	0.327	0.327	<0.001	<0.001
Trt × sub		0.036	0.036	<0.001	<0.001	<0.001	<0.001	0.002	0.002	<0.001	<0.001	0.545	0.545	<0.001	<0.001
Trt × *t*		<0.001	<0.001	<0.001	<0.001	<0.001	<0.001	<0.001	<0.001	<0.001	<0.001	<0.001	<0.001	<0.001	<0.001
Trt × sub × *t*		<0.001	<0.001	<0.001	<0.001	<0.001	<0.001	0.003	0.003	<0.001	<0.001	0.005	0.005	<0.001	<0.001

^1^ HC, High concentrate; HF, High forage; SEM, Standard error of means; Trt, Treatment; Sub, Substrate; *t*, time. ^a–v^ Means with different superscripts within the same column differ, *p* < 0.05.

## Data Availability

Not applicable.

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
