# Peer review of "Effects of Essential Oil Blends on In Vitro Apparent and Truly Degradable Dry Matter, Efficiency of Microbial Production, Total Short-Chain Fatty Acids and Greenhouse Gas Emissions of Two Dairy Cow Diets"

_animals, 2022, doi:10.3390/ani12172185_

Round 1

Reviewer 1 Report

Dear authors,

The manuscript is addressing an interesting topic of research. However, lack of information is observed in some sections. See below a few comments:

1) A justification is needed for testing also different substrates and times.

2) More details are needed in relation to 'Estimation of greenhouse gases'.

3) More references to previous studies conducted by others (focusing mainly on in vitro trials) are needed to complete 'Discussion section' to elucidate the impact of EOBs (1-9), substrates (HF vs. HC) and times (6 vs. 24 h).

4) Conclusions need to be reviewed by considering other research aspects.

Best regards,

Reviewer.

Author Response

Reviewer #1

Dear authors,

The manuscript is addressing an interesting topic of research. However, lack of information is observed in some sections. See below a few comments:

Response

Authors would like to thank the reviewer for giving us the opportunity to revise and improve our manuscript.

 Reviewer #1

1) A justification is needed for testing also different substrates and times.

Response

Authors have added some text to address the concern of the reviewer -  “Additionally, the effects of EOBs tend to be dependent on the diet or substrate, bioactive compounds, incubation time, and level of inclusion, etc.”

 Reviewer #1

2) More details are needed in relation to 'Estimation of greenhouse gases'.

Response

Done.

 Reviewer #1

3) More references to previous studies conducted by others (focusing mainly on in vitro trials) are needed to complete 'Discussion section' to elucidate the impact of EOBs (1-9), substrates (HF vs. HC) and times (6 vs. 24 h).

Response

Authors appreciate the comments from the reviewer but unfortunately, the current study is unique as we used an extensive number of EOBs. To our knowledge, no other study has the number of treatments, substrate and fermentation periods but we did compare our results with several in vitro studies.

 Reviewer #1

4) Conclusions need to be reviewed by considering other research aspects.

Best regards,

Response

Authors have added more text to the Conclusion.

Reviewer 2 Report

Regarding the MS entitled '' Effects of essential oil blends on in vitro apparent and truly degradable dry matter, efficiency of microbial production, total short-chain fatty acids and greenhouse gas emissions of two dairy cow diets'' The main shortage of this study is that this study determined the in vitro analyses only. Moreover, LSD values are strange and even higher than means, please double check.

Abstract

L29. On which basis the authors used these mixtures of essential oils?

L34. Something wrong with this sentence, please revise

More information about the experimental design should be added in the abstract.

The number of replicates is very small and the authors had 9 treatments, this is the main drawback of the current study.

 One conclusion sentence should include the recommendation from this study.

Introduction

The introduction is very weak and very short. the authors should define why they chose these essential oils and refer to other previous published papers in the field?

Hypothesis is missing.

 Materials and Methods

L93-94. Add ref

L98. in vitro, italic throughout the manuscript

All headings and subheadings should be numbered.

L123. (CH4, CO2, H2S and NH3-N), delete from the heading

L130. by [11], define the authors, Anele et al. [11] and also L133.

L159. on the method 8 of ANKOM Technologies analytical methods, please rephrase and define.

Please add Table 1 in the material and methods sections

 Results

L178. The PF was influenced by the EOBs (P<0.001) and substrates (P=0.005), please clarify.

L188. These abbreviations are mentioned before, please revise

L193. P=0.204, non-significant so you do not have to add the p value

L194-195. Please clarify how it was different.

Tables should be added after its specific parts of the results

L252. Table 3, correct

L255 and 265. Correct table number

Discussion

L315. Figures??

Author Response

Reviewer #2

Regarding the MS entitled '' Effects of essential oil blends on in vitro apparent and truly degradable dry matter, efficiency of microbial production, total short-chain fatty acids and greenhouse gas emissions of two dairy cow diets'' The main shortage of this study is that this study determined the in vitro analyses only. Moreover, LSD values are strange and even higher than means, please double check.

Response

Authors would like to thank the reviewer for giving us the opportunity to revise and improve our manuscript. The study is from a project that will end with studies on cannulated dairy cows. The LSD values are from outputs files of SAS analytical package and I believe they are correct. The LSD can be used to compare differences for the treatment x substrate x time interaction.

Reviewer #2

Abstract

L29. On which basis the authors used these mixtures of essential oils?

Response

Authors had a statement in the Materials and Methods under “Essential oil blend” that informs readers our choice of the essential oils. “The EOBs were formulated after an extensive review of previous literature and the percent of each oil in the blend was adjusted based on previous results in literature”.

Reviewer #2

L34. Something wrong with this sentence, please revise

Response

The symbol for micro was not imported correctly. Authors have included the correct symbol and a minor revision.

Reviewer #2

More information about the experimental design should be added in the abstract.

Response

Done.

Reviewer #2

The number of replicates is very small and the authors had 9 treatments, this is the main drawback of the current study.

Response

Authors believe that 6 replicates (3 replicates x 2 runs) are adequate for an in vitro batch culture.  

Reviewer #2

One conclusion sentence should include the recommendation from this study.

Response

Authors have added more text in the conclusion. This is an on-going study which will terminate in an in vivo study, so it is still at a preliminary stage.

Reviewer #2

Introduction

The introduction is very weak and very short. the authors should define why they chose these essential oils and refer to other previous published papers in the field?

Response

Authors have added more text in the Introduction section. Authors also included this statement - The EOBs were formulated after an extensive review of previous literature and the percent of each oil in the blend was adjusted based on previous results in literature.

Reviewer #2

Hypothesis is missing.

Response

Authors have included it.

Reviewer #2

Materials and Methods

L93-94. Add ref

Response

Authors have included a few references.

Reviewer #2

L98. in vitro, italic throughout the manuscript

Response

Done.

Reviewer #2

All headings and subheadings should be numbered.

Response

Done.

 Reviewer #2

L123. (CH4, CO2, H2S and NH3-N), delete from the heading

Response

Done.

Reviewer #2

L130. by [11], define the authors, Anele et al. [11] and also L133.

Response

Done.

Reviewer #2

L159. on the method 8 of ANKOM Technologies analytical methods, please rephrase and define.

Response

Done.

Reviewer #2

Please add Table 1 in the material and methods sections

Response

Done.

Reviewer #2

Results

L178. The PF was influenced by the EOBs (P<0.001) and substrates (P=0.005), please clarify.

Response

Done.

Reviewer #2

L188. These abbreviations are mentioned before, please revise

Response

Authors revised the abbreviations as NDF digestibility (NDFD), etc.  

Reviewer #2

L193. P=0.204, non-significant so you do not have to add the p value

Response

Done.

Reviewer #2

L194-195. Please clarify how it was different.

Response

Done.

Reviewer #2

Tables should be added after its specific parts of the results

Response

Done.

Reviewer #2

L252. Table 3, correct

Response

Yes.

Reviewer #2

L255 and 265. Correct table number

Response

Table numbers are correct.

Reviewer #2

Discussion

L315. Figures??

Response

Authors have replaced “figures” with “numbers”.

Reviewer 3 Report

I include below the comments and suggestions for the authors of this ms:

- It is necessary to include the country in the first affiliation as following authors guidelines provided by the journal.

- Line 34. I suppose the authors would refer to µl instead of litres, the ms should be amended in this sense.

- Line 45. Even that it is a well-known acronym, it should be included between parenthesis Dry matter as the explanation of DM.

- Line 47. The same happens with VFA, Volatile fatty acids should be included between parenthesis.

- The abstract, as stated in guidelines, should be a maximum of 200 words. In my opinion, the information the composition of all EOBs is perfectly described on material and methods, so it could be described in a summarized way to fulfil the guidelines recommendations.

- Keywords usually serve to find the paper by using different words to those included in the paper’s title. In these sense I advise the authors to change those words that have been yet included in the title to help researchers to find the paper when published.

- Line 73. The full name of this acronym IACUC must be included.

- Line 78. It would be interesting to have also collected samples of other types forage as grass silage highly used in dairy cattle nutrition around the world.

- Along the ms, it is used two types of nomenclature for referring the same concept (EOBs), for example EOB2 and EOB 2, with and without space. It should be unified to the same terminology.

- Line 123. The same occurs with CH4, CO2, H2S and NH3 that are expressed in a different way H2 (line 387). All of them have to be expressed using the same model, preferably using subindexes.

- Statistical analysis. In my opinion, in the case of being possible, adding a posthoc analysis when differences were found for treatments, time, etc. would contribute to improve the quality of the results shown.

- Line 166. The full name of this acronym LSD must be included.

- Line 193. The value p=0.204 is referred to NDF but not to ADF and ADL, the values referred particularly to ADF and ADL should be included.

- Line 201. It is said that the concentration of acetate it is increased in all EOB treatments except for EOB 2, I suppose it is a mistake and the authors would like to write EOB 1, in that sense ms should be amended.

- In the results of table 5, from my point of view it is better to follow the same order of VFA than in the table in order to help the reader to follow the results. So the order of explained butyrate and isobutyrate should be the same of the table.

- Tables 1 and 3. The tables should be susceptible of being understood without more information, so, in order to help the comprehension of all the words used DM acronym should be included in a footnote.  

- Tables 2, 3, 4 and 5. In the same sense, the significance of EOBs should be included in a footnote. From my point of view, the tables should be easy to understand if in the column treatment instead of 1, 2, 3, … it  would be written EOB 1, EOB 2, EOB 3, …

- Tables 3, 4 and 5. The same happens with 6 and 24, in the case of time. I think it would be easier to understand if it is included 6 hours and 24 hours. Also in these tables, for the interactions Trtxt and Trtxsubxt, it is not explained that t is time in the footnotes, it should be included.

- Tables 3, 4 and 5 are identified as 1, 2 and 3. They must be correctly identified.

- Conclusions. It is said that differences were observed by using EOBs on in vitro fermentation and that these additives could be used to improve livestock production. I would be grateful if the authors could explain better/widespread in which aspects they could help to improve livestock production.

- References. Style of references should be arranged as stated in the guidelines.

E.g. Journal Articles:
1. Author 1, A.B.; Author 2, C.D. Title of the article. 
Abbreviated Journal Name YearVolume, page range.

Author Response

Reviewer #3

I include below the comments and suggestions for the authors of this ms:

Response

Authors would like to thank the reviewer for giving us the opportunity to revise and improve our manuscript.

Reviewer #3

- It is necessary to include the country in the first affiliation as following authors guidelines provided by the journal.

Response

Done.

Reviewer #3

- Line 34. I suppose the authors would refer to µl instead of litres, the ms should be amended in this sense.

Response

Done. That was a typo.

Reviewer #3

- Line 45. Even that it is a well-known acronym, it should be included between parenthesis Dry matter as the explanation of DM.

Response

The sentence was deleted in the revied manuscript.

Reviewer #3

- Line 47. The same happens with VFA, Volatile fatty acids should be included between parenthesis.

Response

Done.

Reviewer #3

- The abstract, as stated in guidelines, should be a maximum of 200 words. In my opinion, the information the composition of all EOBs is perfectly described on material and methods, so it could be described in a summarized way to fulfil the guidelines recommendations.

Response

Authors have deleted information on the composition of all EOBs and other sentences.

Reviewer #3

- Keywords usually serve to find the paper by using different words to those included in the paper’s title. In these sense I advise the authors to change those words that have been yet included in the title to help researchers to find the paper when published.

Response

Done.

Reviewer #3

- Line 73. The full name of this acronym IACUC must be included.

Response

Done.

Reviewer #3

- Line 78. It would be interesting to have also collected samples of other types forage as grass silage highly used in dairy cattle nutrition around the world.

Response

Authors will consider the reviewer’s suggestion in future studies.

Reviewer #3

- Along the ms, it is used two types of nomenclature for referring the same concept (EOBs), for example EOB2 and EOB 2, with and without space. It should be unified to the same terminology.

Response

Authors deleted the space to ensure consistence.

Reviewer #3

- Line 123. The same occurs with CH4, CO2, H2S and NH3 that are expressed in a different way H2 (line 387). All of them have to be expressed using the same model, preferably using subindexes.

Response

Done.

Reviewer #3

- Statistical analysis. In my opinion, in the case of being possible, adding a posthoc analysis when differences were found for treatments, time, etc. would contribute to improve the quality of the results shown.

Response

Authors agree with the reviewer on adding posthoc analysis in the Tables but due to the number of variables (treatment, substrate and time) in the current study, using the LSD method is the simplest way to check for differences in the Tables. Including the superscripts will make it cumbersome to interpret the results.

Reviewer #3

- Line 166. The full name of this acronym LSD must be included.

Response

Done.

Reviewer #3

- Line 193. The value p=0.204 is referred to NDF but not to ADF and ADL, the values referred particularly to ADF and ADL should be included.

Response

Based on the recommendation of one of the reviewers, the P-value was deleted.

Reviewer #3

- Line 201. It is said that the concentration of acetate it is increased in all EOB treatments except for EOB 2, I suppose it is a mistake and the authors would like to write EOB 1, in that sense ms should be amended.

Response

Authors would like to thank the reviewer for pointing this error.

Reviewer #3

- In the results of table 5, from my point of view it is better to follow the same order of VFA than in the table in order to help the reader to follow the results. So the order of explained butyrate and isobutyrate should be the same of the table.

Response

Done.

Reviewer #3

- Tables 1 and 3. The tables should be susceptible of being understood without more information, so, in order to help the comprehension of all the words used DM acronym should be included in a footnote.  

Response

Done.

Reviewer #3

- Tables 2, 3, 4 and 5. In the same sense, the significance of EOBs should be included in a footnote. From my point of view, the tables should be easy to understand if in the column treatment instead of 1, 2, 3, … it  would be written EOB 1, EOB 2, EOB 3, …

Response

Done.

Reviewer #3

- Tables 3, 4 and 5. The same happens with 6 and 24, in the case of time. I think it would be easier to understand if it is included 6 hours and 24 hours. Also in these tables, for the interactions Trtxt and Trtxsubxt, it is not explained that t is time in the footnotes, it should be included.

Response

Done.

Reviewer #3

- Tables 3, 4 and 5 are identified as 1, 2 and 3. They must be correctly identified.

Response

Done.

Reviewer #3

- Conclusions. It is said that differences were observed by using EOBs on in vitro fermentation and that these additives could be used to improve livestock production. I would be grateful if the authors could explain better/widespread in which aspects they could help to improve livestock production.

Response

Authors included additional text to highlight their ability to reduce GHG and improve efficiency of microbial production.

Reviewer #3

- References. Style of references should be arranged as stated in the guidelines.

E.g. Journal Articles:
1. Author 1, A.B.; Author 2, C.D. Title of the article. Abbreviated Journal Name YearVolume, page range.

Response

Done.

Round 2

Reviewer 2 Report

Regarding the manuscript entitled '' Effects of essential oil blends on in vitro apparent and truly degradable dry matter, efficiency of microbial production, total short-chain fatty acids and greenhouse gas emissions of two dairy cow diets''

Thank you for revisions.

Yet, some comments are not addressed.

Again, LSD0.05 or LSD0.01 values should be less than the means of the groups. If LSD is larger than the means, how the authors compare between means of groups. If so, it means that there is no significant difference between groups or time. I recommended that authors to double-check their data input files that may be have units varied from that presented in tables. Otherwise, differences between treatment means should be determined by Tukey’s test and SEM is enough.

Again the tables numbers are in correct.

Some means in the tables have negative values, please clarify or double check. IVADDM, Gas (ml/g DM), Carbon dioxide (mg/g DM), and Ammonia (mmol/g DM), and HydroS (mmol/g DM).

Author Response

Reviewer #2

Again, LSD0.05 or LSD0.01 values should be less than the means of the groups. If LSD is larger than the means, how the authors compare between means of groups. If so, it means that there is no significant difference between groups or time. I recommended that authors to double-check their data input files that may be have units varied from that presented in tables. Otherwise, differences between treatment means should be determined by Tukey’s test and SEM is enough.

Response

Authors have fixed the issue with the LSD values. The LSD values in the revised manuscript are significantly lower than previous values.

Reviewer #2

Again the tables numbers are in correct.

Response

Authors deleted the following text (and total volatile fatty acids (TVFA)) which may have resulted to the confusion on Table numbers. Please see deleted text on L200.

Reviewer #2

Some means in the tables have negative values, please clarify or double check. IVADDM, Gas (ml/g DM), Carbon dioxide (mg/g DM), and Ammonia (mmol/g DM), and HydroS (mmol/g DM).

Response

Total gas produced by the treatments were subtracted from total gas produced in the blanks (without any substrate or diet). The implication is the blanks produced more gas than these treatments and this is expected due to the suppressing effects of some of the EOBs. The negative numbers are valid.